# Online Linear Optimization with Many Hints

**Aditya Bhaskara**
Department of Computer Science
University of Utah
Salt Lake City, UT
bhaskaraaditya@gmail.com

**Ashok Cutkosky**
Dept. of Electrical and Computer Engineering
Boston University
Boston, MA
ashok@cutkosky.com

**Ravi Kumar**
Google Research
Mountain View, CA
ravi.k53@gmail.com

**Manish Purohit**
Google Research
Mountain View, CA
mpurohit@google.com

## Abstract

We study an online linear optimization (OLO) problem in which the learner is provided access to $K$ "hint" vectors in each round prior to making a decision. In this setting, we devise an algorithm that obtains logarithmic regret whenever there exists a convex combination of the $K$ hints that has positive correlation with the cost vectors. This significantly extends prior work that considered only the case $K = 1$. To accomplish this, we develop a way to combine many arbitrary OLO algorithms to obtain regret only a logarithmically worse factor than the minimum regret of the original algorithms in hindsight; this result is of independent interest.

## 1 Introduction

In this paper we consider a variant of the classic online linear optimization (OLO) problem [28]. In OLO, at each time step, an algorithm must play a point $x_t$ in some convex set $X \subseteq \mathbb{R}^d$, and then it is presented with a cost vector $c_t$ and incurs loss $\langle c_t, x_t \rangle$. This process repeats for $T$ time steps. The algorithm's performance is measured via the *regret* relative to some comparison point $u \in X$, defined as $\sum_{t=1}^{T} \langle c_t, x_t - u \rangle$.

This problem is of fundamental interest in a variety of fields. OLO algorithms are directly applicable for solving the learning with expert advice problem as well as online convex optimization [4]. Further, in machine learning, one frequently encounters stochastic convex optimization problems, which may be solved via online convex optimization through the online-to-batch conversion [3]. Many of the popular optimization algorithms used in machine learning practice today (e.g., [10, 18]) can be analyzed within the OLO framework. For more details and further applications, we refer the interested reader to the excellent texts [4, 12, 24].

OLO is well-understood from an algorithmic viewpoint. For the vanilla version of the problem, algorithms with regret $O(\sqrt{T})$ are known [28, 16] and this bound is tight [4]. An interesting line of research has been to identify situations and conditions where the regret can be substantially smaller than $\sqrt{T}$. Towards this, Dekel et al. [9] proposed the study of OLO augmented with hints; their work was motivated by an earlier work of Hazan and Megiddo [14]. In their setup, the algorithm has access to a hint at each time step before it responds and this hint is guaranteed to be more than $\alpha$-correlated with the cost vector. They obtained an algorithm with a regret of $O(d/\alpha \cdot \log T)$, where $d$ is the dimension of the space. Very recently, Bhaskara et al. [2] generalized their results to the case when the hints can be arbitrary, i.e., not necessarily weakly positively correlated at each time step. They

obtain an algorithm with a dimension-free regret bound of roughly $O(\sqrt{B}/\alpha \cdot \log T)$, where $B$ is the number of (bad) time steps when the hints are less than $\alpha$-correlated with the cost vector.

While this line of work gives a promising way to go beyond $\sqrt{T}$ regret, in many situations, it is not clear how to obtain a hint sequence that correlates well with the cost vector in most time steps. Prior work on optimism [13, 23, 26] has suggested using costs from earlier time steps, costs from earlier batches, or even from other learning algorithms. This suggests that it is often possible to obtain *multiple* sources that provide hint sequences, and we may hope that an appropriate combination of them correlates well with the cost vector in most time steps.

In this work, we focus on this natural setting in which multiple (arbitrary) hints are available to the algorithm at each time step. If some aggregate of the hints is helpful, we would like to perform as well as if we knew this aggregate a priori. As we discuss in Section 3.2, this is difficult because the benefit of aggregating multiple hints is a nonlinear function of the benefits of the individual hints. Even if all the hints are individually bad, an algorithm may be able to gain significantly from using some convex combination of the hints.

**Our results.** Let $K$ be the number of hints available at each time step. We obtain an online learning algorithm for the constrained case, where the responses of the algorithm must be inside the unit ball. Our algorithm obtains a regret of roughly $O(\sqrt{B/\alpha \cdot \log T} + (\log T + \sqrt{(\log T)(\log K)})/\alpha)$, where $B$ is the number of time steps when the *best* convex combination of the hints is less than $\alpha$-correlated with the cost vector. We refer to Theorem 5 for the formal guarantees. We also obtain lower bounds showing the dependence of the regret on both $K$ and $\alpha$ is essentially tight (Section 3.3).

Our algorithm is designed in two stages. In the first stage, we assume that the optimal threshold $\alpha$ is known. We build an algorithm based on carefully defining a *smoothed* hinge loss function that captures the performance over the entire simplex of hints and then using Mirror Descent on the losses. The second stage eliminates the assumption on knowing $\alpha$ by developing a new *combiner* algorithm. This is a general randomized procedure that combines a collection of online learning algorithms and achieves regret only logarithmically worse than the minimum regret of the original algorithms. (This combiner is of independent interest and we show a few applications outside our main theme.)

For the unconstrained setting (defined formally below), we develop an algorithm that achieves a (relative) regret of roughly $O(\log T \cdot (\sqrt{B/\alpha} + \sqrt{\log K}/\alpha))$, where $B$ is once again defined as before. Our algorithm thus competes with the best convex combination of the hints.

## 2 Preliminaries

Let $[T] = \{1, \ldots, T\}$. In the classical online learning setting, at each time $t \in [T]$, an algorithm $\mathcal{A}$ responds with a vector $x_t \in \mathbb{R}^d$. *After* the response, a cost vector $c_t \in \mathbb{R}^d$ is revealed and the algorithm incurs a cost of $\langle c_t, x_t \rangle$. We assume that $\|c_t\| \leq 1$, $\forall t \leq T$, where $\| \cdot \|$ always indicates the $\ell_2$-norm unless specified otherwise. The *regret* of the algorithm $\mathcal{A}$ for a vector $u \in \mathbb{R}^d$ is

$$\mathcal{R}_\mathcal{A}(u, \vec{c}) = \mathcal{R}_\mathcal{A}(u, \vec{c}, T) = \sum_{t=1}^{T} \langle c_t, x_t - u \rangle.$$

A *hint* is a vector $h \in \mathbb{R}^d$, $\|h\| \leq 1$ and $\vec{h} = (h_1, h_2, \ldots)$ is a sequence of hints. We consider the case when there are *multiple* hints available to the algorithm $\mathcal{A}$. In each round $t$, the algorithm $\mathcal{A}$ gets $K$ hints $h_t^{(1)}, \ldots, h_t^{(K)}$ *before* it responds with $x_t$. While some of the hint sequences might be good and others might be misleading, our goal is to design an algorithm that does nearly as well as if we were just given the best sequence of hints. Let $H = \{\vec{h}^{(1)}, \ldots, \vec{h}^{(K)}\}$ denote the set of hint sequences. The regret definition is the same as always and is denoted $\mathcal{R}_\mathcal{A}(u, \vec{c} \mid H)$.

Let $\Delta_K \subset \mathbb{R}^K$ denote the simplex. Given a sequence $\vec{w} = (w_1, w_2, \ldots)$ of vectors in $\Delta_K$, we write $H(\vec{w})$ to indicate the sequence of hints with $t$th hint $\sum_{i=1}^{K} w_t^{(i)} \cdot h_t^{(i)}$, where $w_t^{(i)}$ indicates the $i$th component of $w_t$. If $\vec{w}$ is a constant sequence $(w, w, \ldots)$, then we write $H(w)$ instead of $H(\vec{w})$.

Let $\alpha > 0$ be a fixed *threshold*. For a fixed hint sequence $\vec{h}$, we define $B_\alpha^{\vec{h}}$ to be the set of all time steps where the hint $h_t$ is *bad*, i.e., less than $\alpha$-correlated with the cost $c_t$. Formally, we have

$$B_\alpha^{\vec{h}} = \left\{ t \in [T] : \langle c_t, h_t \rangle < \alpha \cdot \|c_t\|^2 \right\}.$$

We consider two settings to measure the worst-case regret of an algorithm. In the *constrained* setting, we are given some set $\mathcal{B}$ and the worst-case regret of $\mathcal{A}$ is defined as $\mathcal{R}_{\mathcal{A}}(\mathcal{B}, \vec{c} \mid H) = \sup_{u \in \mathcal{B}} \mathcal{R}_{\mathcal{A}}(u, \vec{c} \mid H)$; in this paper we take $\mathcal{B} = \{x \in \mathbb{R}^d : \|x\| \leq 1\}$, the unit ball. In the *unconstrained* setting, the regret of $\mathcal{A}$ is measured over $u \in \mathbb{R}^d$ and we denote it by $\mathcal{R}_{\mathcal{A}}(u, \vec{c} \mid H)$, which we will bound uniformly by another function of $u$.

## 2.1 Single hint case

Now we recall and mildly improve the results of [2] for the case that there is a *single* hint at every time step (i.e., $K = 1$). We will consider the case of fixed and *known* $\alpha$; note that the algorithm of [2] is agnostic to $\alpha$, but we show that by committing to a fixed $\alpha$ we can improve the regret bound. We will remove this dependence on a known $\alpha$ later in Section 4. The modification to both the algorithm and the analysis is not hard, and so we defer the proof to Appendix A.

**Theorem 1.** *For any $0 < \alpha < 1$, there exists an algorithm $1$-$\mathrm{HINT}_\alpha$ that runs in $O(d)$ time per update, takes a single hint sequence $\vec{h}$, and guarantees regret:*

$$
\mathcal{R}_{1\text{-}\mathrm{HINT}_\alpha}(\mathcal{B}, \vec{c} \mid \{\vec{h}\}) \leq \frac{1}{2} + 4 \left( \sqrt{\sum_{t \in B_\alpha^{\vec{h}}} \|c_t\|^2} + \frac{\log T}{\alpha} + 2\sqrt{\frac{(\log T) \sum_{t=1}^{T} \max(0, -\langle c_t, h_t \rangle)}{\alpha}} \right)
$$

$$
\leq O \left( \sqrt{\frac{(\log T)|B_\alpha^{\vec{h}}|}{\alpha}} + \frac{\log T}{\alpha} \right).
$$

In contrast, the bound in [2] had the factor $(\log T)/\alpha$ instead of $\sqrt{(\log T)/\alpha}$ (in the first term).

# 3 Constrained setting: Known $\alpha$

Recall that in the constrained setting, the algorithm must always respond with $x_t \in \mathcal{B}$, the unit ball. Our main result is a version of Theorem 1 for $K > 1$, and it will extend the previous works of [2, 9]. The high-level approach is quite natural: we design a *meta-learner* that maintains a loss for each hint sequence at each time, and at time $t$, uses the losses to decide on an appropriate convex combination $w_t$ of the hints $\{h_t^i\}_{i=1}^{K}$. We then run an instance of the single hint algorithm, $1$-$\mathrm{HINT}_\alpha$, using this combination as the provided hint.

There are two main challenges with this approach. First, the regret bound of $1$-$\mathrm{HINT}_\alpha$ depends on the quantity $B_\alpha^{H(\vec{w})}$, which depends on the convex combination $w_t$ used at each step $t$, and it is not clear how to relate it to the corresponding terms for the individual hint sequences. Second, the regret bound assumes a knowledge of $\alpha$, while our final goal is to compete with the best possible (unknown) $\alpha$. We deal with the second challenge in Section 4 by designing a general combination algorithm. In this section we address the first challenge; all the algorithms in this section assume a fixed and known value of $\alpha$. Any omitted proofs can be found in Appendix B.

## 3.1 Multiplicative weights on hint sequences

We first show a result weaker than the main result of this section (Theorem 5). The algorithm is conceptually simpler, and it demonstrates what one obtains by using a simple multiplicative weight update (MWU) rule to learn the best hint sequence among the $K$ sequences, and then use Theorem 1 with the learned hint sequence. Since the single-hint regret bound (Theorem 1) depends on just the number of time steps when the hint has a poor correlation with the cost vector, using an MWU algorithm using binary losses suffices. In particular, if $\vec{h}^{\mathrm{MW}}$ denotes the hint sequence obtained from the multiplicative weights algorithm, we can show that $|B_\alpha^{\vec{h}^{\mathrm{MW}}}| \leq O(\min_{i \in K} |B_\alpha^{\vec{h}^{(i)}}|)$. We defer the proof of the following theorem to Appendix B.1.

**Theorem 2.** *Let $\alpha \in (0,1)$ be given. There exists a randomized algorithm $\mathcal{A}_{MW}$ for OLO with $K$ hint sequences that has a regret bound of*

$$\mathbb{E}[\mathcal{R}_{\mathcal{A}_{MW}}(\mathcal{B}, \vec{c} \mid H)] \leq O\left(\inf_{i \in K} \sqrt{\frac{(\log T)(|B_\alpha^{\vec{h}^{(i)}}| + \log K)}{\alpha}} + \frac{\log T}{\alpha}\right).$$

Note that this is usually weaker than Theorem 5 because it competes only with the best *individual* hint sequence, and not necessarily the best *convex combination* of hints. It can only be a better bound if $K \gg T$ so that $\log K = \omega(\log T)$.

## 3.2 Smoothed hinge loss

The multiplicative weights approach allows us to obtain regret guarantees that depend on the number of bad hints in the best of the $K$ hint sequences. But, what we would really like is for the regret bound to scale with the number of bad hints in the best *convex combination* of the hint sequences. This can be a significant gain: consider the setting in which $K = 2$ and $\alpha = \frac{1}{4}$, and on even iterations $t$ we have $\langle c_t, h_t^{(1)} \rangle = -1/4$ while on odd iterations $\langle c_t, h_t^{(1)} \rangle = 1$. Suppose $h_t^{(2)}$ is the same, but has high correlation on even iterations and negative correlation on odd iterations. Then both $h_t^{(1)}$ and $h_t^{(2)}$ have $T/2$ "bad hints", but the convex combination $\frac{h_t^{(1)}}{2} + \frac{h_t^{(2)}}{2}$ has *no* bad hints! This highlights the fundamental problem with the multiplicative weights approach: linear combinations of hints might result in much better performance than the corresponding linear combination of the respective performances of the hints.

---

**Algorithm 1** $K$-HINTS$_\alpha$

**Input:** Parameter $\alpha$
  Define $\psi(w) = (\log K) + \sum_{i=1}^{K} w^{(i)}(\log w^{(i)})$

  Initialize 1-HINT$_{\alpha/2}$
  Initialize $w_1 \leftarrow (1/K, \ldots, 1/K) \in \Delta_K$
  **for** $t = 1, \ldots, T$ **do**
    Get hints $h_t^{(1)}, \ldots, h_t^{(K)}$
    Send $h_t \leftarrow \sum_{i=1}^{K} w_t^{(i)} h_t^{(i)}$ to 1-HINT$_{\alpha/2}$
    Get $x_t$ from 1-HINT$_{\alpha/2}$
    Respond $x_t$, receive cost $c_t$
    Send $c_t$ to 1-HINT$_{\alpha/2}$
    $\ell_t(w) \leftarrow \ell\left(\langle c_t, \sum_{i=1}^{K} w^{(i)} h_t^{(i)} \rangle, \alpha\|c_t\|^2\right)$
    $g_t \leftarrow \nabla \ell_t(w_t)$
    $w_{t+1} \leftarrow \text{argmin}_{w \in \Delta_K}$
      $\langle g_{1:t}, w \rangle + \sqrt{\frac{(\log K) + \sum_{\tau=1}^{t} \|g_\tau\|_\infty^2}{\log K}} \psi(w)$
  **end for**

---

We will address this issue by considering a specially crafted loss function that more accurately captures performance over the entire simplex of hints. Intuitively, we would like to design a loss function such that for any $w \in \Delta_K$, the loss $\ell_t(w)$ is low if and only if $h_t(w) = \sum_{i=1}^{K} h_t^{(i)} w^{(i)}$ has the desired correlation with $\|c_t\|^2$. Once we have the appropriate loss function, we can then use an online learning algorithm on the losses $\ell_t$ to obtain the desired convex combination of hints at each time step.

Formally, the following smoothed version of the hinge loss is adequate for our purposes.

$$\ell(a,b) = \begin{cases} 0 & a > b \\ \frac{1}{b}(b-a)^2 & a \in [0,b] \\ b - 2a & a < 0 \end{cases} \tag{1}$$

For any $w \in \Delta_K$, we define the loss function as $\ell_t(w) = \ell(\langle c_t, h_t(w) \rangle, \alpha\|c_t\|^2)$ where $h_t(w) = \sum_{i=1}^{K} w^{(i)} h_t^{(i)}$ and $\ell(\cdot)$ is as defined in (1). We first present several important properties of this loss function in the following proposition.

**Proposition 3.** *Let $\alpha \in (0,1)$ be fixed and for all $t \in [T]$, let $\ell_t(w) = \ell(\langle c_t, h_t(w) \rangle, \alpha\|c_t\|^2)$. Then,*

*(a). $\ell_t$ is convex and non-negative.*
*(b). If $h_t(w)$ is $\alpha$-good (i.e., $\langle c_t, h_t(w) \rangle \geq \alpha\|c_t\|^2$), then $\ell_t(w) = 0$ and $0 \in \partial \ell_t(w)$.*
*(c). If $h_t(w)$ is not $(\alpha/2)$-good (i.e., $\langle c_t, h_t(w) \rangle < \alpha\|c_t\|^2/2$), then $\ell_t(w) \geq \alpha\|c_t\|^2/4$.*
*(d). $\ell_t$ is 2-Lipschitz with respect to the $\ell_1$-norm.*
*(e). $\|\nabla \ell_t(w)\|_\infty^2 \leq \frac{4}{\alpha} \ell_t(w)$ for all $w \in \Delta_K$.*

*(f).* $\ell_t(w) \le \alpha \|c_t\|^2 + 2\max(0, -\langle c_t, h_t(w)\rangle)$.

*Proof.* Properties (a)–(b) are immediate from the definition of $\ell(\cdot, \cdot)$. For property (c), if $\langle c_t, h_t(w)\rangle < 0$, then we have $\ell_t(w) = \alpha\|c_t\|^2 - 2\langle c_t, h_t(w)\rangle \ge \alpha\|c_t\|^2$. On the other hand, if $0 \le \langle c_t, h_t(w)\rangle < \alpha\|c_t\|^2/2$, then we have $\ell_t(w) = \left(\frac{1}{\alpha\|c_t\|^2}\right)\left(\alpha\|c_t\|^2 - \langle c_t, h_t(w)\rangle\right)^2 \ge \alpha\|c_t\|^2/4$.

For the next properties, define $f : \mathbb{R} \to \mathbb{R}$ by $f(x) = \ell(x, \alpha\|c_t\|^2)$. By manually computing derivatives of $f$ we can see that $f$ is 2-Lipschitz and 1-smooth. Further since $|\langle c_t, h_t^{(i)}\rangle| \le 1$ for all $i$, we have that $g(w) = \langle c_t, h_t(w)\rangle$ is 1-Lipschitz with respect to the $\ell_1$-norm. Therefore $\ell_t$ must be 2-Lipschitz with respect to the $\ell_1$-norm, proving (d).

By inspecting the derivatives of $f$, we see that $f'(x)^2 \le \frac{4}{\alpha\|c_t\|^2} f(x)$. Further, we have $\nabla\ell_t(w)^{(i)} = \langle c_t, h_t^{(i)}\rangle f'(\langle c_t, h_t(w)\rangle)$. Therefore $\|\nabla\ell_t(w)\|_\infty \le \|c_t\| f'(\langle c_t, h_t(w)\rangle)$, from which (e) follows. For (f), we observe that $f(x) \le \alpha\|c_t\|^2 + 2\max(0, -x)$. $\qquad\square$

We are now ready to present our final algorithm $K$-HINTS$_\alpha$. At each timestep $t$, we first choose a $w_t \in \Delta_K$ via FTRL using an entropic regularizer on the losses $\ell_t$ (see last line of Algorithm 1). We then supply the learned hint $h_t(w_t) = \sum_{i=1}^{K} w_t^{(i)} h_t^{(i)}$ to an instance of the single hint algorithm. For technical reasons, we use the single hint algorithm 1-HINT$_{\alpha/2}$ where the desired correlation with the cost vector is set to $\alpha/2$ instead of $\alpha$. Algorithm 1 presents the complete pseudocode. The performance of the FTRL subroutine can be bounded via classical results in FTRL (see [19]) used in concert with the smoothness of the losses $\ell_t$, following [25]. The final result is the following Proposition 4, which we prove in Appendix B.

**Proposition 4.** *Let $w_t \in \Delta_K$ be chosen via FTRL on the losses $\ell_t$ as in Algorithm 1. Then, for any $w_\star \in \Delta_K$, we have*

$$\sum_{t=1}^{T} \ell_t(w_t) \le \frac{22\log K}{\alpha} + 2\sum_{t=1}^{T} \ell_t(w_\star).$$

With this proposition, we can prove the main result of this section:

**Theorem 5.** *Let $\alpha \in (0,1)$ be given. Then $K$-HINTS$_\alpha$ on OLO with $K$ hint sequences guarantees:*

$$\mathcal{R}_{K\text{-HINTS}_\alpha}(\mathcal{B}, \vec{c} \mid H) \le O\left(\inf_{w\in\Delta_K} \sqrt{(\log T)\sum_{t\in B_\alpha^{H(w)}} \|c_t\|^2} + \sqrt{\frac{(\log T)\sum_{t=1}^{T}\max(0, -\langle c_t, h_t(w)\rangle)}{\alpha}}\right.$$

$$\left. + \frac{(\log T) + \sqrt{(\log T)(\log K)}}{\alpha}\right)$$

$$\le O\left(\inf_{w\in\Delta_K} \sqrt{\frac{(\log T)|B_\alpha^{H(w)}|}{\alpha}} + \frac{(\log T) + \sqrt{(\log T)(\log K)}}{\alpha}\right).$$

*In the above, $h_t(w) = \sum_{i=1}^{K} w^{(i)} h_t^{(i)}$ is the $t$th hint of the sequence $H(w)$ for $w \in \Delta_K$.*

*Proof.* Let $w_\star$ be an arbitrary element of $\Delta_K$. By Proposition 3(f), we have $\ell_t(w_\star) \le \alpha\|c_t\|^2 + 2\max(0, -\langle c_t, h_t(w_\star)\rangle)$ for all $t$, and $\ell_t(w_\star) = 0$ if $\langle c_t, h_t(w_\star)\rangle \ge \alpha\|c_t\|^2$. Therefore,

$$\sum_{t=1}^{T} \ell_t(w_\star) \le \sum_{t\in B_\alpha^{H(w_\star)}} \left(\alpha\|c_t\|^2 + 2\max(0, -\langle c_t, h_t(w_\star)\rangle)\right) = Q, \tag{2}$$

where we have defined the variable $Q = \sum_{t\in B_\alpha^{H(w_\star)}} \alpha\|c_t\|^2 + 2\max(0, -\langle c_t, h_t(w_\star)\rangle)$.

Further, by definition of the smoothed hinge loss, we have $\ell_t(w_t) \ge \max(0, -\langle c_t, h_t(w_t)\rangle)$ for all $t \in [T]$. Therefore, by Proposition 4 and (2), we have

$$\sum_{t=1}^{T} \max(0, -\langle c_t, h_t(w_t)\rangle) \le \sum_{t=1}^{T} \ell_t(w_t) \le 2Q + \frac{22\log K}{\alpha}. \tag{3}$$

Also, since the loss function is always non-negative, we have

$$\sum_{t=1}^{T}\ell_t(w_t) \geq \sum_{t\in B_{\alpha/2}^{H(\vec{w})}}\ell_t(w_t) \geq \sum_{t\in B_{\alpha/2}^{H(\vec{w})}}\frac{\alpha\|c_t\|^2}{4},$$

where the second inequality uses Proposition 3(c). Once again, using Proposition 4 and (2), we have

$$\sum_{t\in B_{\alpha/2}^{H(\vec{w})}}\|c_t\|^2 \leq \frac{8Q}{\alpha} + \frac{88\log K}{\alpha^2}. \tag{4}$$

Finally, recall that we have sent the hint sequence $H(\vec{w}) = (h_1(w_1),\ldots,h_T(w_T))$ to the algorithm 1-$\text{HINT}_{\alpha/2}$. Thus by Theorem 1, we have:

$$\mathcal{R}_{K\text{-HINTS}_\alpha}(\mathcal{B},\vec{c}\mid H) \leq \frac{1}{2} + 4\left(\sqrt{\sum_{t\in B_{\alpha/2}^{H(\vec{w})}}\|c_t\|^2} + \frac{\log T}{\alpha} + \sqrt{\frac{(2\log T)\sum_{t=1}^{T}\max(0,-\langle c_t, h_t(w_t)\rangle)}{\alpha}}\right)$$

substituting (3) and (4),

$$\leq \frac{1}{2} + 4\left(\sqrt{\frac{8Q}{\alpha} + \frac{88\log K}{\alpha^2}} + \frac{\log T}{\alpha} + \sqrt{\frac{2(\log T)\left(2Q + \frac{22\log K}{\alpha}\right)}{\alpha}}\right). \tag{5}$$

The final result now follows from the definition of $Q$ and simple calculations. $\qquad\square$

**Non-negatively correlated hints.** Recall that in the case of $K = 1$, [9] obtains a regret of $O((\log T)/\alpha)$ in the case where *all* the hints are $\alpha$-correlated with $c_t$. A weaker assumption is to have $\langle h_t, c_t\rangle \geq 0$ at all steps, with the $\alpha$-correlation property holding at all but $B_\alpha$ time steps. In this case, [2] showed that the regret must be at least $\Omega(\sqrt{B_\alpha})$, and also gave an algorithm that achieves a regret of $O\left(\sqrt{B_\alpha} + \frac{\log T}{\alpha}\right)$. Using Theorem 5, we obtain this bound for general $K$.

**Corollary 6.** *Consider OLO with $K$ hint sequences where for every $t$ and every hint $h_t^{(i)}$, we have the property that $\langle h_t^{(i)}, c_t\rangle \geq 0$. Further, suppose that for some $\alpha > 0$, there exists an (unknown) convex combination $w$ such that for the hint sequence $H(w)$, the number of hints that do not satisfy $\langle h_t(w), c_t\rangle \geq \alpha\|c_t\|^2$ is at most $B_\alpha$. Then there exists an algorithm that achieves a regret at most*

$$O\left(\sqrt{B_\alpha} + \frac{\log T + \sqrt{\log K}}{\alpha}\right).$$

Specifically, before substituting to obtain (5), observe that under the non-negative correlation assumption, $\max(0,\langle c_t, h_t(w_t)\rangle) = 0$ for all $t$, and thus we only have the first two terms of (5). This gives the desired bound.

## 3.3 Lower bounds

In this section we provide some lower bounds, focusing on the dependence on $K$ and $\alpha$. Our primary technique is to specify hint sequences and costs such that, even given the hint, the cost is $\alpha$-correlated with some combination of hints, but otherwise is a random variable with mean 0 and variance 1. The high variance in the costs guarantees nearly $\sqrt{T}$ regret, which we express in terms of $\alpha$ and $K$ to achieve our bounds. Omitted proofs can be found in Appendix C. We begin with a lower bound showing that the dependence on $\sqrt{(\log K)/\alpha}$ holds even in one dimension.

**Theorem 7.** *For any $\alpha$ and $T \geq \frac{1}{\alpha}\log\frac{1}{\alpha}$, there exists a sequence $\vec{c}$ of costs and a set $H$ of hint sequences, $|H| = K$ for some $K$, such that: (i) there is a convex combination of the $K$ hints that always has correlation $\alpha$ with the costs and (ii) the regret of any online algorithm is at least $\sqrt{\frac{\log K}{2\alpha}}$.*

Next, we show that a dependence on $1/\alpha$ is also unavoidable:

**Theorem 8.** *In the two-dimensional constrained setting, there is a sequence $\vec{h}$ and $\vec{c}$ of hints and costs $(K = 1)$ such that: (i) $\forall t$, $\langle h_t, c_t \rangle \geq \alpha$, and (ii) the regret of any online algorithm is at least $\Omega(1/\alpha)$.*

Together, these bounds show that the $\sqrt{\log K}$ and $1/\alpha$ terms in our upper bounds are necessary. There is a gap in our upper and lower bounds in terms of the dependence on $\log T$, and the gap between $\frac{\sqrt{\log K}}{\alpha}$ and $\max\{\sqrt{\log K}, 1/\alpha\}$.

## 4 Combining learners

In Section 3, we presented an algorithm for online learning with multiple hints. However, the algorithm required knowing $\alpha$, the desired correlation between a hint $h$ and the cost vector $c_t$. In this section, we eliminate this assumption. To do this, we design a generic way to combine incomparable-in-foresight regret guarantees obtained by different algorithms and essentially get the best regret among them in hindsight. With this combiner, handling unknown $\alpha$ is easy: consider $K$-$\text{HINTS}_\alpha$ for different values of $\alpha$ and apply the combiner to get the best among them.

The results in this section apply in the constrained setting and to both the hints and the classical no-hints case (see [5] for analogous results that apply in the unconstrained setting when the base algorithms are "parameter-free"). These combiner algorithms themselves are of independent interest and lead to other applications in the constrained online learning setting that we elaborate in Appendix E.

---

**Algorithm 2** Deterministic combiner $\mathcal{C}_{\text{det}}$.

**Input:** Online algorithms $\mathcal{A}_1, \ldots, \mathcal{A}_K$
Reset $\mathcal{A}_1$
Set $i \leftarrow 1, \gamma \leftarrow 1, r \leftarrow 0, \tau \leftarrow 1, r_0^{i,\gamma} \leftarrow 0$
**for** $t = 1, \ldots, T$ **do**
   Get $y_\tau$ from $\mathcal{A}_i$ and respond $x_t \leftarrow y_\tau$
   Get cost $c_t$, define $g_\tau \leftarrow c_t$
   Send $g_\tau$ to $\mathcal{A}_i$ as $\tau$th cost
   Set $r_\tau^{i,\gamma} \leftarrow \sup_{u \in \mathcal{B}} \sum_{\tau'=1}^{\tau} \langle g_{\tau'}, y_{\tau'} - u \rangle$
   **if** $r_\tau^{i,\gamma} > \gamma$ **then**
     **if** $i = K$ **then**
       Set $\gamma \leftarrow 2\gamma$
     **end if**
     Set $i \leftarrow (i \bmod K) + 1$
     Set $\tau \leftarrow 1$
     Set $r_0^{i,\gamma} \leftarrow 0$
     Reset $\mathcal{A}_i$
   **end if**
   Set $\tau \leftarrow \tau + 1$
**end for**

---

For technical reasons, we need the following definition of a "monotone regret bound". Essentially all regret bounds known for online linear optimization satisfy this definition.

**Definition 9** (Monotone regret bound). *An online learning algorithm $\mathcal{A}$ is associated with a* monotone regret bound *$\mathcal{S}([a,b], \vec{c})$, if $\mathcal{S}(\cdot, \cdot)$ is such that when $\mathcal{A}$ is run on only the costs $c_a, \ldots, c_b$, producing outputs $x_a, \ldots, x_b$, we have the guarantee:*

$$\sup_{u \in \mathcal{B}} \sum_{t=a}^{b} \langle c_t, x_t - u \rangle \leq \mathcal{S}([a,b], \vec{c}),$$

*and further it satisfies $\mathcal{S}([a', b'], \vec{c}) \leq \mathcal{S}([a,b], \vec{c})$ for all sequences $\vec{c}$ whenever $[a', b'] \subseteq [a, b]$.*

Note that if an algorithm $\mathcal{A}$ has a monotone regret bound $\mathcal{S}(\cdot, \cdot)$, then the total regret experienced by algorithm $\mathcal{A}$ is bounded as $\mathcal{R}_\mathcal{A}(\mathcal{B}, \vec{c}) \leq \mathcal{S}([1, T], \vec{c})$.

### 4.1 Deterministic combiner

We first design a simple deterministic algorithm $\mathcal{C}_{\text{det}}$ that combines $K$ online learning algorithms with monotone regret bounds and obtains a regret that is at most $K$ times the regret suffered by the best algorithm on any given cost sequence. The combiner starts with an initial guess of the regret $\gamma$ and guesses that the first algorithm is the best, playing its predictions. It keeps trusting the current choice of the best algorithm until the regret it incurs exceeds the current guess $\gamma$; once that happens, it chooses the next algorithm. Once all the algorithms have been tried, it doubles the guess $\gamma$ and starts over. Notice that this does not require knowledge of the bounds $\mathcal{S}_i$; these can be replaced with the "true" regret bounds, rather than simply the best bound that present analysis is capable of delivering.

**Theorem 10.** *Suppose $\mathcal{A}_1, \ldots, \mathcal{A}_K$ are deterministic OLO algorithms that are associated with monotone regret bounds $\mathcal{S}_1, \ldots, \mathcal{S}_K$. Suppose $\forall t$, $\sup_{x,y\in\mathcal{B}}\langle c_t, x-y\rangle \le 1$. Then, we have:*

$$\mathcal{R}_{\mathcal{C}_{\det}}(\mathcal{B}, \vec{c}) \le K\left(4 + 4\min_i \mathcal{S}_i([1,T], \vec{c})\right).$$

*Proof sketch.* We give a brief sketch here and defer the formal proof to Appendix D. We can divide the operation of Algorithm 2 into phases in which $\gamma$ is constant. In each phase, Algorithm 2 incurs a regret of at most $\gamma + 1$ from each of the $K$ algorithms for a total regret of at most $K(\gamma+1)$. Let $P$ denote the total number of phases and let $j = \arg\min_i \mathcal{S}_i([1,T], \vec{c})$ be the algorithm with the least total regret. In the $(P-1)$th phase, algorithm $A_j$ must have incurred a regret of at least $2^{P-2}$ (otherwise we would not have the $P$th phase). Since we assume that $\mathcal{S}_j$ is a monotone regret bound, it follows that $\min_i \mathcal{S}_i([1,T], \vec{c}) \ge 2^{P-2}$ and hence $P \le \max(1, 2 + \log_2(\min_i \mathcal{S}_i([1,T], \vec{c})))$. Since $\gamma = 2^{p-1}$ in phase $p$, we can bound the total regret incurred by Algorithm 2 as

$$\sup_{u\in\mathcal{B}}\sum_{t=1}^{T}\langle c_t, x_t - u\rangle \le \sum_{p=1}^{P} K(2^{p-1}+1) \le K(P + 2^P) \le K2^{P+1}$$

$$\le K\left(4 + 4\min_i \mathcal{S}_i([1,T], \vec{c})\right). \qquad \square$$

## 4.2 Randomized combiner

The deterministic combiner $\mathcal{C}_{\det}$, while achieving the best regret among $\mathcal{A}_1, \ldots, \mathcal{A}_K$, incurs a factor $K$. We now show that using randomization, this factor can be made $O(\log K)$ in expectation.

Intuitively, $\mathcal{C}_{\det}$ incurs the factor $K$ since it might be unlucky and have to cycle through all the $K$ algorithms even after it correctly guesses $\gamma$. We can avoid this worst-case behavior by selecting the base algorithm uniformly at random, rather than in a deterministic order. We formally describe this randomized combiner $\mathcal{C}_{\text{rand}}$ in Algorithm 4 in Appendix D. Informally, in each phase with constant $\gamma$, at each time step, $\mathcal{C}_{\text{rand}}$ simulates all the $K$ algorithms and maintains a candidate set $C$ of algorithms that have incurred a regret of at most $\gamma$. Once the current algorithm incurs a regret of $\ge \gamma$, $\mathcal{C}_{\text{rand}}$ selects the next algorithm to be one from the set $C$ uniformly at random. Suppose the algorithms in $C$ are ranked by the first time they incur a regret bound of $\gamma$. Since an algorithm $\mathcal{A}_i$ is chosen uniformly at random, in expectation, by the time $\mathcal{A}_i$ incurs a regret of $\gamma$, half of the algorithms in $C$ have already incurred at least $\gamma$ regret and thus the size of $C$ halves at each step. Thus, we can argue that we only cycle through $O(\log K)$ base algorithms in each phase. We defer the formal proof of the following theorem to Appendix D.

**Theorem 11.** *Suppose $\mathcal{A}_1, \ldots, \mathcal{A}_K$ are deterministic OLO algorithms with monotone regret bounds $\mathcal{S}_1, \ldots, \mathcal{S}_K$. Suppose for all $t$, $\sup_{x,y\in\mathcal{B}}\langle c_t, x-y\rangle \le 1$. Then for any fixed sequence $\vec{c}$ of costs (i.e., an oblivious adversary), we have:*

$$\mathbb{E}\left[\mathcal{R}_{\mathcal{C}_{\text{rand}}}(\mathcal{B}, \vec{c})\right] \le \log_2(K+1) \cdot \left(4 + 4\min_i \mathcal{S}_i([1,T], \vec{c})\right).$$

*Further, if $\vec{c}$ is allowed to depend on the algorithm's randomness (i.e., an adaptive adversary), then*

$$\mathcal{R}_{\mathcal{C}_{\text{rand}}}(\mathcal{B}, \vec{c}) \le K\left(4 + 4\min_i \mathcal{S}_i([1,T], \vec{c})\right).$$

## 4.3 Constrained setting: Unknown $\alpha$

For any fixed $\alpha > 0$, Theorem 5 yields a monotone regret bound. For $1 \le i \le \log T$, let $\mathcal{A}_i$ denote the instantiation of Algorithm 1 with $\alpha_i = 2^{-i}$. By Theorem 5, each algorithm $\mathcal{A}_i$ is associated with a monotone regret bound $\mathcal{S}_i(\cdot, \cdot)$ such that

$$\mathcal{R}_{\mathcal{A}_i}(\mathcal{B}, \vec{c}) \le \mathcal{S}_i([1,T], \vec{c}) = O\left(\inf_{w\in\Delta_K}\sqrt{\frac{(\log T)|B_{\alpha_i}^{H(w)}|}{\alpha_i}} + \frac{(\log T) + \sqrt{(\log T)(\log K)}}{\alpha_i}\right).$$

Further since $|B_{\alpha_{i+1}}^{H(w)}| \le |B_{\alpha_i}^{H(w)}|$, we have $\mathcal{S}_{i+1}(\cdot, \vec{c}) \le 2\mathcal{S}_i(\cdot, \vec{c})$. Applying Theorem 11 on these $\log T$ algorithms thus yields the following result.

**Theorem 12.** *Given a set $H = \{\vec{h}^1, \ldots, \vec{h}^K\}$ of hint sequences, there exists a randomized algorithm $\mathcal{A}$ such that for any fixed sequence of cost vectors $\vec{c}$, the expected regret $\mathbb{E}[\mathcal{R}_{\mathcal{A}}(\mathcal{B}, \vec{c} \mid H)]$ is at most:*

$$O\left(\inf_{\alpha}\inf_{w\in\Delta_K}\left\{(\log\log T)\cdot\left(\sqrt{\frac{(\log T)|B_\alpha^{H(w)}|}{\alpha}}+\frac{(\log T)+\sqrt{(\log T)(\log K)}}{\alpha}\right)\right\}\right).$$

## 5 Unconstrained setting

In this section, we develop an algorithm that leverages multiple hints in the unconstrained setting. Recall that in this setting, the output $x_t$ and comparison point $u$ are allowed to range over all of $\mathbb{R}^d$. Thus we cannot hope to bound regret by a uniform constant for all $u$. Instead, we bound the regret as a function of $\|u\|$. This setting has seen increased interest [6, 7, 11, 21, 22], and recently the notion of hints has also been studied [2, 5]. Here, we consider multiple hints in the unconstrained setting. Unlike the constrained case, this algorithm does not need to know $\alpha$ and hence does not need the combiner. The algorithm again competes with the best convex combination of the hints.

Following [2, 5], our algorithm initializes $K+1$ unconstrained online learners. The first online learner ignores the hints and attempts to output $x_t$ to minimize the regret. Each of the following $K$ online learners is restricted to output real numbers $y_t^{(i)}$ for $i=1,\dots,K$ rather than points in $\mathbb{R}^d$. The final output of our algorithm is then given by $\hat{x}_t = x_t + \sum_{i=1}^{K} y_t^{(i)} h_t^{(i)}$. Intuitively, the $i$th one-dimensional algorithm is attempting to learn how "useful" the $i$th hint sequence is. Upon receiving the cost $c_t$, we provide the $i$th one-dimensional algorithm with the cost $\langle c_t, h_t^{(i)}\rangle$. Note that we are leaning heavily on the lack of constraints in this construction. Our regret bound is given in Theorem 13, proved in Appendix F.

**Theorem 13.** *There is an algorithm $\mathcal{A}$ for the unconstrained setting such that for any $u \in \mathbb{R}^d$ and any $\alpha \in (0,1)$, we have*

$$\mathcal{R}_\mathcal{A}(u,\vec{c}\mid H) = O\left(\inf_{w\in\Delta_K}\left\{\|u\|(\log T)\left(\frac{\sqrt{\log K}}{\alpha}+\sqrt{\frac{B_\alpha^{H(w)}}{\alpha}}\right)\right\}\right).$$

## 6 Conclusions

In this paper we obtained algorithms for online linear optimization in the presence of many hints that can be imperfect. Besides generalizing previous results on online optimization with hints, our contributions include a simple algorithm for combining arbitrary learners that seems to have broader applications. Interesting future research directions include tightening the dependence on $\alpha$ in various cases and exploring the possibility of improved bounds for specific online optimization problems.

## Broader Impact

Our work focuses on theoretical foundations. Online learning methods have had direct impact in domains such as online advertising. But primarily, the methods developed are used in improving other optimization procedures, and thus only have an indirect impact. We believe that there are no adverse ethical aspects or potentially negative societal consequences of our work.

## Acknowledgments and Disclosure of Funding

Aditya Bhaskara is partially supported by NSF (CCF-2008688) and by a Google Faculty Research Award.

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
