[Supplementary Material]

# A  Single hint setting

In this section, we modify the construction of [2] in the single hint setting to take into account knowledge of the parameter $\alpha$. Our goal is to prove Theorem 1. The algorithm is nearly identical to that of [2] and most of the analysis is the same. We refer the reader to the original reference for complete details.

---

**Algorithm 3** 1-HINT$_\alpha$

---

**Input:** Parameter $\alpha$
    Define $\lambda_0 = 1$ and $r_0 = 1$
    Set procedure $\mathcal{A}$ to be Algorithm 2 in [2].
    **for** $t = 1, \ldots, T$ **do**
        Get hint $h_t$
        Get $\overline{x}_t$ from procedure $\mathcal{A}$, and set

$$x_t \leftarrow \overline{x}_t + \frac{(\|\overline{x}_t\|^2 - 1)}{2r_t}h_t$$

        Play $x_t$ and receive cost $c_t$
        Set $r_{t+1} \leftarrow \sqrt{r_t^2 + \frac{\alpha \max(0, -\langle c_t, h_t \rangle)}{\log(T)}}$
        Define $\sigma_t = \frac{|\langle c_t, h_t \rangle|}{r_t}$
        Define $\lambda_t$ as the solution to:

$$\lambda_t = \frac{\|c_t\|^2}{\sum_{\tau=1}^{t} \sigma_\tau + \lambda_\tau}$$

        Define the loss $\ell_t(x) = \langle c_t, x \rangle + \frac{|\langle c_t, h_t \rangle|}{2r_t}(\|x\|^2 - 1)$. Send the loss function $\ell_t$ to $\mathcal{A}$
    **end for**

---

The only difference between our algorithm 1-HINT$_\alpha$ and Algorithm 1 of [2] is the definition of $r_t$: when we set $r_{t+1} = \sqrt{r_t^2 + \frac{\max(0, -\langle c_t, h_t \rangle)\alpha}{\log(T)}}$, [2] instead sets $r_{t+1} = \sqrt{r_t^2 + \max(0, -\langle c_t, h_t \rangle)}$. We can now prove Theorem 1, which we restate below for reference:

**Theorem 1.** *For any $0 < \alpha < 1$, there exists an algorithm 1-HINT$_\alpha$ that runs in $O(d)$ time per update, takes a single hint sequence $\vec{h}$, and guarantees regret:*

$$\mathcal{R}_{1\text{-HINT}_\alpha}(\mathcal{B}, \vec{c} \mid \{\vec{h}\}) \leq \frac{1}{2} + 4\left(\sqrt{\sum_{t \in B_\alpha^{\vec{h}}} \|c_t\|^2} + \frac{\log T}{\alpha} + 2\sqrt{\frac{(\log T)\sum_{t=1}^{T} \max(0, -\langle c_t, h_t \rangle)}{\alpha}}\right)$$

$$\leq O\left(\sqrt{\frac{(\log T)|B_\alpha^{\vec{h}}|}{\alpha}} + \frac{\log T}{\alpha}\right).$$

*Proof.* Following [2], we observe that since $\mathcal{A}$ always returns $\overline{x}_t \in \mathcal{B}$, $x_t \in \mathcal{B}$. Further,

$$\langle c_t, x_t - u \rangle \leq \ell_t(x_t) - \ell_t(u) + \frac{\max(0, -\langle c_t, h_t \rangle)}{r_t},$$

and $\ell_t$ is $\sigma_t$-strongly convex.

Next, by [2] Lemma 3.4, we have

$$\mathcal{R}_{1\text{-HINT}_\alpha}(\mathcal{B}, \vec{c} \mid \{\vec{h}\}) \leq \sum_{t=1}^{T} \frac{\max(0, -\langle c_t, h_t \rangle)}{r_t} + \sum_{t=1}^{T} \ell_t(\overline{x}_t) - \ell_t(u).$$

We can bound the first sum as:

$$\sum_{t=1}^{T} \frac{\max(0, -\langle c_t, h_t \rangle)}{r_t} \leq \frac{\log T}{\alpha} \sum_{t=1}^{T} \frac{\alpha \max(0, -\langle c_t, h_t \rangle)/\log T}{r_t}$$

$$\leq \frac{2 \log T}{\alpha} \sqrt{\sum_{t=1}^{T} \frac{\alpha \max(0, -\langle c_t, h_t \rangle)}{\log T}}$$

$$\leq \sqrt{2 \frac{\sum_{t=1}^{T} (\log T) \max(0, -\langle c_t, h_t \rangle)}{\alpha}}.$$

For the second sum, we appeal to Lemma 3.6 of [2], which yields:

$$\sum_{t=1}^{T} \ell_t(\bar{x}_t) - \ell_t(u) \leq \frac{1}{2} + 4 \left( \sqrt{\sum_{t \in B_\alpha^{\vec{h}}} \|c_t\|^2} + \frac{r_T (\log T)}{\alpha} \right)$$

$$\leq \frac{1}{2} + 4 \left( \sqrt{\sum_{t \in B_\alpha^{\vec{h}}} \|c_t\|^2} + \frac{\sqrt{(\log^2 T) + (\log T) \alpha \sum_{t=1}^{T} \max(0, -\langle c_t, h_t \rangle)}}{\alpha} \right)$$

$$\leq \frac{1}{2} + 4 \left( \sqrt{\sum_{t \in B_\alpha^{\vec{h}}} \|c_t\|^2} + \frac{\log T}{\alpha} + \sqrt{\frac{(\log T) \sum_{t=1}^{T} \max(0, -\langle c_t, h_t \rangle)}{\alpha}} \right).$$

Combining these identities now yields the desired theorem. $\square$

# B Full proofs: Constrained setting

## B.1 Proof of Theorem 2

**Theorem 2.** *Let $\alpha \in (0, 1)$ be given. There exists a randomized algorithm $\mathcal{A}_{MW}$ for OLO with $K$ hint sequences that has a regret bound of*

$$\mathbb{E}[\mathcal{R}_{\mathcal{A}_{MW}}(\mathcal{B}, \vec{c} \mid H)] \leq O \left( \inf_{i \in K} \sqrt{\frac{(\log T)(|B_\alpha^{\vec{h}^{(i)}}| + \log K)}{\alpha}} + \frac{\log T}{\alpha} \right).$$

*Proof.* At each time step $t$, our goal is to pick a single hint $h_t \in \{h_t^{(1)}, \ldots, h_t^{(K)}\}$. We instantiate this problem as an instance of the standard prediction with $K$ experts problem with binary losses defined as follows.

$$\ell_{t,i} = \begin{cases} 0 & \text{if } |\langle c_t, h_t^{(i)} \rangle| \geq \alpha \|c_t\|, \\ 1 & \text{otherwise.} \end{cases}$$

Let $\vec{h}^{(i^*)}$ denote the hint sequence with minimum loss in hindsight, i.e., $i^* = \operatorname{argmin}_{i \in K} \sum_t \ell_{t,i}$. We note that by definition of the losses $\ell$, we have $\sum_t \ell_{t,i^*} = |B_\alpha^{\vec{h}^{(i^*)}}|$. Let $\vec{h}^{\text{MW}} = (h_1^{(i_1)}, h_2^{(i_2)}, \ldots)$ be the sequence of hints obtained by running the classical Multiplicative Weights algorithm with a decay factor of $\eta = \frac{1}{2}$. Then by standard analysis (e.g., Theorem 2.1 of Arora et al. [1]), we have the following.

$$\mathbb{E}[\sum_t (\ell_{t,i_t} - \ell_{t,i^*})] \leq 2 \log K + \frac{1}{2} \sum_t (\ell_{t,i^*}). \tag{6}$$

Substituting $|B_\alpha^{\vec{h}^{(i^*)}}| = \sum_t \ell_{t,i^*}$ and rearranging,

$$\mathbb{E}[|B_\alpha^{\vec{h}^{\text{MW}}}|] = \mathbb{E}[\sum_t \ell_{t,i_t}] \leq \frac{3}{2} |B_\alpha^{\vec{h}^{(i^*)}}| + 2 \log K. \tag{7}$$

We then run an instance of the single hint algorithm, 1-$\text{HINT}_\alpha$, with the hint sequence $\vec{h}^{\text{MW}}$. Applying Theorem 1 yields the following.

$$\mathbb{E}[\mathcal{R}_{A_{\text{MW}}}(\mathcal{B}, \vec{c} \mid H)] \leq O\left(\mathbb{E}\left[\sqrt{\frac{(\log T)|B_\alpha^{\vec{h}^{\text{MW}}}|}{\alpha}}\right] + \frac{\log T}{\alpha}\right)$$

$$\leq O\left(\sqrt{\frac{(\log T)\mathbb{E}\left[|B_\alpha^{\vec{h}^{\text{MW}}}|\right]}{\alpha}} + \frac{\log T}{\alpha}\right)$$

$$\leq O\left(\sqrt{\frac{(\log T)(|B_\alpha^{\vec{h}^{(i^*)}}| + \log K)}{\alpha}} + \frac{\log T}{\alpha}\right),$$

where the first inequality follows from Jensen's inequality and the second one follows from (7). □

## B.2 Proof of Proposition 4

Before proving Proposition 4, we apply the analysis of adaptive follow-the-regularized-leader (FTRL) as in [19] to obtain:

**Proposition 14.** *For any $w_\star \in \Delta_K$, we have:*

$$\sum_{t=1}^T (\ell_t(w_t) - \ell_t(w_\star)) \leq 2\sqrt{(\log^2 K) + (\log K)\sum_{t=1}^T \|g_t\|_\infty^2}.$$

*Proof.* To begin, recall that the entropic regularizer $\psi(w) = \log(K) + \sum_{i=1}^K w^{(i)}(\log w^{(i)})$ is 1-strongly-convex with respect to the 1-norm over $\Delta_K$, has minimum value 0 and maximum value $\log K$.

Then, standard bounds for FTRL (e.g., [19, Theorem 1]) tell us that:

$$\sum_{t=1}^T \ell_t(w_t) - \ell_t(w_\star) \leq \sqrt{\frac{(\log K) + \sum_{t=1}^T \|g_t\|_\infty^2}{\log K}} \psi(w_\star) + \sum_{t=1}^T \frac{\|g_t\|_\infty^2 \sqrt{\log K}}{2\sqrt{(\log K) + \sum_{\tau=1}^{t-1} \|g_\tau\|_\infty^2}}$$

$$\leq \sqrt{\frac{(\log K) + \sum_{t=1}^T \|g_t\|_\infty^2}{\log K}} \psi(w_\star) + \sum_{t=1}^T \frac{\|g_t\|_\infty^2 \sqrt{\log K}}{2\sqrt{\sum_{\tau=1}^{t} \|g_\tau\|_\infty^2}}$$

$$\leq \sqrt{\frac{(\log K) + \sum_{t=1}^T \|g_t\|_\infty^2}{\log K}} \psi(w_\star) + \sqrt{(\log K)\sum_{t=1}^T \|g_t\|_\infty^2}$$

$$\leq 2\sqrt{(\log^2 K) + (\log K)\sum_{t=1}^T \|g_t\|_\infty^2}.$$

□

Now with Proposition 14 in hand, we can restate and prove:

**Proposition 4.** *Let $w_t \in \Delta_K$ be chosen via FTRL on the losses $\ell_t$ as in Algorithm 1. Then, for any $w_\star \in \Delta_K$, we have*

$$\sum_{t=1}^T \ell_t(w_t) \leq \frac{22 \log K}{\alpha} + 2\sum_{t=1}^T \ell_t(w_\star).$$

*Proof.* From Proposition 3, we have

$$\sum_{t=1}^{T} \|g_t\|_{\infty}^2 \le \sum_{t=1}^{T} \frac{4}{\alpha} \ell_t(w_t).$$

Combining this with the regret bound of Proposition 14 yields:

$$\sum_{t=1}^{T} \ell_t(w_t) - \ell_t(w_\star) \le 2\sqrt{(\log^2 K) + \frac{4 \log K}{\alpha} \sum_{t=1}^{T} \ell_t(w_t)}.$$

If we set $R = \sum_{t=1}^{T} \ell_t(w_t) - \ell_t(w_\star)$, we can rewrite the above as:

$$R \le 2\sqrt{(\log^2 K) + \frac{4 \log K}{\alpha} R + \frac{4 \log K}{\alpha} \sum_{t=1}^{T} \ell_t(w_\star)}.$$

Now we use $\sqrt{a+b} \le \sqrt{a} + \sqrt{b}$ and solve for $R$:

$$R \le \frac{16 \log K}{\alpha} + \sqrt{4 \log^2 K + \frac{16 \log K}{\alpha} \sum_{t=1}^{T} \ell_t(w_\star)}$$

$$\le \frac{18 \log K}{\alpha} + \sqrt{\frac{16 \log K}{\alpha} \sum_{t=1}^{T} \ell_t(w_\star)}$$

$$\implies \sum_{t=1}^{T} \ell_t(w_t) \le \sum_{t=1}^{T} \ell_t(w_\star) + \frac{18 \log K}{\alpha} + \sqrt{\frac{16 \log K}{\alpha} \sum_{t=1}^{T} \ell_t(w_\star)}.$$

Next, observe that $\sqrt{aX} \le X + \frac{a}{4}$ for all $a, X \ge 0$, so that

$$\sum_{t=1}^{T} \ell_t(w_t) \le 2 \sum_{t=1}^{T} \ell_t(w_\star) + \frac{22 \log K}{\alpha}.$$

as desired. □

## C  Lower bound proofs

**Theorem 7.** *For any $\alpha$ and $T \ge \frac{1}{\alpha} \log \frac{1}{\alpha}$, there exists a sequence $\vec{c}$ of costs and a set $H$ of hint sequences, $|H| = K$ for some $K$, such that: (i) there is a convex combination of the $K$ hints that always has correlation $\alpha$ with the costs and (ii) the regret of any online algorithm is at least $\sqrt{\frac{\log K}{2\alpha}}$.*

*Proof.* Consider a one-dimensional problem with $K = \frac{T 2^B}{B}$ hint sequences for $B = \alpha T$. Suppose $T \ge \frac{\log(1/\alpha)}{\alpha}$, so that $2^B \ge \frac{T}{B}$ and $\log K \le 2B = 2T\alpha$. We group the hint sequences into $\frac{T}{B}$ groups each of size $2^B$. We now specify the hint sequence in the $i$th such group for some arbitrary $i$. All hints in the $i$th group are 0 for all $t \notin [(i-1)B, iB-1]$ and for $t \in [iB, (i+1)B)$, the hints take on the $2^B$ possible sequences of $\pm 1$. Then it is clear that for *any* sequence of $\pm 1$ costs, there is a convex combination of hints that places weight $B/T$ on exactly one hint sequence in each of the $T/B$ groups such that the linear combination always has correlation $\alpha = B/T$ with the cost.

Let the costs be random $\pm 1$, so that the expected regret is $\sqrt{T}$. Then we conclude by observing $\sqrt{\log K}/\sqrt{2\alpha} \le \sqrt{2\alpha T}/\sqrt{2\alpha} = \sqrt{T}$. □

**Theorem 8.** *In the two-dimensional constrained setting, there is a sequence $\vec{h}$ and $\vec{c}$ of hints and costs $(K = 1)$ such that: (i) $\forall t$, $\langle h_t, c_t \rangle \ge \alpha$, and (ii) the regret of any online algorithm is at least $\Omega(1/\alpha)$.*

*Proof.* Let $e_0$ and $e_1$ be orthogonal unit vectors, and let $h_t = e_0$ for all $t$. Suppose that $c_t = \alpha e_0 \pm \sqrt{1 - \alpha^2} e_1$ for all $t$, where the sign is chosen uniformly at random. Note that any online algorithm has expected reward at most $\alpha T$ (since it cannot gain anything in the $e_1$ direction, so it is best to place all the mass along $e_0$).

On the other hand, we have

$$\mathbb{E}\left[\left\|\sum_{t=1}^{T} c_t\right\|^2\right] = \alpha^2 T^2 + T(1 - \alpha^2),$$

and thus the optimal vector in hindsight achieves a reward $\sqrt{\alpha^2 T^2 + T(1 - \alpha^2)}$. Thus the regret is

$$\frac{T(1 - \alpha^2)}{\alpha T + \sqrt{\alpha^2 T^2 + T(1 - \alpha^2)}} \geq \frac{T(1 - \alpha^2)}{2\alpha T + \sqrt{T(1 - \alpha^2)}} \geq \frac{1}{\alpha},$$

for sufficiently large $T$. $\qquad\square$

## D  Proofs from Section 4

**Theorem 10.** *Suppose $\mathcal{A}_1, \ldots, \mathcal{A}_K$ are deterministic OLO algorithms that are associated with monotone regret bounds $\mathcal{S}_1, \ldots, \mathcal{S}_K$. Suppose $\forall t$, $\sup_{x,y \in \mathcal{B}} \langle c_t, x - y \rangle \leq 1$. Then, we have:*

$$\mathcal{R}_{\mathcal{C}_{\text{det}}}(\mathcal{B}, \vec{c}) \leq K\left(4 + 4 \min_i \mathcal{S}_i([1, T], \vec{c})\right).$$

*Proof.* We can divide the operation of Algorithm 2 into phases in which $\gamma$ is constant. Each phase may be further subdivided into sub-phases in which $i$ is constant. First, let us bound the regret in a single phase with fixed $\gamma$. Suppose this phase has $N \leq K$ sub-phases[1]. Let $t_1, \ldots, t_N$ be the time indices at which each sub-phase begins, and let $t_{N+1} - 1$ be the last time index belonging to this phase. Notice that for all $i \leq N$, we must have $r^{i,\gamma}_{t_{i+1} - t_i - 1} \leq \gamma$ since the $i$th sub-phase lasts for $t_{i+1} - t_i$ iterations. Then since $\sup_{x,y} \langle c_{t_{i+1} - 1}, x - y \rangle \leq 1$ for all $i$ and $x, y \in X$, we have $r^{i,\gamma}_{t_{i+1} - t_i} \leq r^{i,\gamma}_{t_{i+1} - t_i - 1} + 1 \leq \gamma + 1$. Now we can write the regret incurred over this phase as:

$$\sup_{u \in X} \sum_{t=t_1}^{t_{N+1} - 1} \langle c_t, x_t - u \rangle \leq \sum_{i=1}^{N} \sup_{u \in X} \sum_{t=t_i}^{t_{i+1} - 1} \langle c_t, x_t - u \rangle \leq \sum_{i=1}^{N} r^{i,\gamma}_{t_{i+1} - t_i} \leq N(\gamma + 1) \leq K\gamma + K.$$

Let $P$ denote the total number of phases. We now show that $P \leq 2 + \max(-1, \log_2(\min_i \mathcal{S}_i([1, T], \vec{c})))$. Suppose otherwise. Let $j = \arg\min_i \mathcal{S}_i([1, T], \vec{c})$ be the algorithm with the least total regret. Let us consider the $(P-1)$th phase. In this phase, $\gamma = 2^{P-2}$. Since $P > 2 + \log_2(\min_i \mathcal{S}_i([1, T], \vec{c}))$, we must have $\min_i \mathcal{S}_i([1, T], \vec{c}) < \gamma$. Consider the $j$th sub-phase in this phase. Since $\gamma$ will eventually increase, this sub-phase must eventually end. Therefore there must be some $t$ and $\tau$ such that $t + \tau < T$ and

$$\sup_{u \in X} \sum_{\tau'=1}^{\tau} \langle c_{t+\tau'}, w_{\tau'} - u \rangle > \gamma,$$

where $w_{\tau'}$ is the output of $\mathcal{A}_j$ after seeing input $c_t, \ldots, c_{t+\tau'-1}$. By the increasing property of $R_j$, we also have:

$$\sup_{u \in X} \sum_{\tau'=1}^{\tau} \langle c_{t+\tau'}, w_{\tau'} - u \rangle \leq \mathcal{S}_j([t, t+\tau], \vec{c}) \leq \mathcal{S}_j([1, T], \vec{c}) < \gamma.$$

which is a contradiction. Therefore $P \leq 2 + \max(-1, \log_2(\min_i \mathcal{S}_i([1, T], \vec{c})))$.

Now we are in a position to calculate the total regret. Let $1 = T_1, \ldots, T_P$ be the start times of the $P$ phases, and let $T_{P+1} - 1 = T$ for notational convenience. Then we have:

$$\sup_{u \in X} \sum_{t=1}^{T} \langle c_t, x_t - u \rangle \leq \sum_{e=1}^{P} \sup_{u \in X} \sum_{t=T_e}^{T_{e+1} - 1} \langle c_t, x_t - u \rangle.$$

Now since the regret in an phase is at most $K\gamma + K$, and $\gamma$ doubles every phase,

$$\leq \sum_{e=1}^{P} K2^{e-1} + K \leq KP + K2^P$$

$$\leq K2^{P+1}$$

$$\leq K\left(4 + 4\min_i \mathcal{S}_i([1,T],\vec{c})\right),$$

where the second-to-last inequality follows from $x \leq 2^x$ for $x \geq 1$, and the last inequality is from case analysis. □

---

**Algorithm 4** Randomized combiner.

---

**Input:** Online algorithms $\mathcal{A}_1, \ldots, \mathcal{A}_K$
Reset $\mathcal{A}_1$
Set $\gamma \leftarrow 1, \tau \leftarrow 1$
Initialize the candidate indices $C \leftarrow [K]$
Choose index $i$ uniformly at random from $C$
**for** $t = 1, \ldots, T$ **do**
  **for** $j \in C$ **do**
    Get $y_\tau^j$, the $\tau$th output of $\mathcal{A}_j$
  **end for**
  Respond $x_t \leftarrow y_\tau^i$
  Get cost $c_t$, define $g_\tau \leftarrow c_t$
  **for** $j \in C$ **do**
    Send $g_\tau$ to $\mathcal{A}_j$ as $\tau$th cost
    Set $r_\tau^{j,\gamma} \leftarrow \sup_{u \in \mathcal{B}} \sum_{\tau'=1}^{\tau} \langle g_{\tau'}, y_{\tau'}^j - u \rangle$
    **if** $r_\tau^{j,\gamma} > \gamma$ **then**
      Set $C \leftarrow C \setminus \{j\}$
    **end if**
  **end for**
  **if** $i \notin C$ **then**
    **if** $C = \emptyset$ **then**
      Set $C \leftarrow [K]$
      Set $\gamma \leftarrow 2\gamma$
    **end if**
    Set $\tau \leftarrow 1$
    Reset $\mathcal{A}_j$ for all $j \in C$
    Select index $i$ uniformly at random from $C$
  **end if**
  Set $\tau \leftarrow \tau + 1$
**end for**

---

**Theorem 11.** *Suppose $\mathcal{A}_1, \ldots, \mathcal{A}_K$ are deterministic OLO algorithms with monotone regret bounds $\mathcal{S}_1, \ldots, \mathcal{S}_K$. Suppose for all $t$, $\sup_{x,y \in \mathcal{B}} \langle c_t, x - y \rangle \leq 1$. Then for any fixed sequence $\vec{c}$ of costs (i.e., an oblivious adversary), Algorithm 4 guarantees:*
$$\mathbb{E}\left[\mathcal{R}_{\mathcal{C}_{\mathrm{rand}}}(\mathcal{B}, \vec{c})\right] \leq \log_2(K+1) \cdot \left(4 + 4\min_i \mathcal{S}_i([1,T],\vec{c})\right).$$

*Further, if $\vec{c}$ is allowed to depend on the algorithm's randomness (i.e., an adaptive adversary), then*
$$\mathcal{R}_{\mathcal{C}_{\mathrm{rand}}}(\mathcal{B}, \vec{c}) \leq K\left(4 + 4\min_i \mathcal{S}_i([1,T],\vec{c})\right).$$

*Proof.* We divide the operation of Algorithm 4 into phases in which $\gamma$ is constant. Each phase is further subdivided into sub-phases in which $i$ is constant. First, let us fix an phase $e$ with a fixed value of $\gamma$ and bound the expected regret incurred in this phase. Let $N$ denote the number of sub-phases in this phase. Just as in the proof of Theorem 10, we can show that the total regret incurred in this phase is at most $N(\gamma + 1)$. However, while there are exactly $K$ sub-phases in any phase of Algorithm 2

(except perhaps the last one), the number of sub-phases in any phase of Algorithm 4 is a random variable.

We now bound $\mathbb{E}[N]$, the expected number of sub-phases in any phase. For the fixed phase $e$, for any time index $t$, let $F(i,t)$ be the smallest index $\tau \geq t$ such that $\sup_{u \in X} \sum_{\tau'=t}^{\tau} \langle c_{\tau'}, w^i(t,\tau') - u \rangle > \gamma$, where we define $w^i(t,\tau')$ to be the output of $A_i$ after seeing input $c_t, \ldots, c_{\tau'-1}$ and $w^i(t,t)$ to be the initial output of $A_i$. We set $F(i,t) = T$ if no such index $\tau \leq T$ exists. Intuitively, $F(i,t)$ denotes the index $\tau \geq t$ when the regret experienced by algorithm $A_i$ that is initialized at time $t$ first exceeds $\gamma$.

Let $C(S,t)$ be the expected number of sub-phases (counting the current one) left in the phase if a sub-phase starts at time $t$ with the specified set of active indices $S$. We define $C(S, T+1) = C(\emptyset, t) = 0$ for all $S$ and $t$ for notational convenience. Note that $C(S,T) = 1$ for all $S$. Further, by definition, we have $\mathbb{E}[N] = C(\{1, 2, \ldots, K\}, t)$ for some $t$ (corresponding to the start of the phase). We claim that $C$ satisfies:

$$C(S,t) = 1 + \frac{1}{|S|} \sum_{i \in S} C(S \setminus \{j \in S \mid F(j,t) \leq F(i,t)\}, F(i,t) + 1).$$

To see this, observe that each index $i \in S$ is equally likely to be selected for the fixed $i$ throughout the sub-phase starting at time $t$. By definition of $F$, the sub-phase will end at time $F(i,t)$ if the selected index is $i$. Further, at the end of the sub-phase, $S$ will be $S \setminus \{j \in S \mid F(j,t) \leq F(i,t)\}$. Therefore, conditioned on selecting index $i$ for this sub-phase, the expected number of sub-phases is $1 + C(S \setminus \{j \in S \mid F(j,t) \leq F(i,t)\}, F(i,t) + 1)$. Since each index is selected with probability $1/|S|$, the stated identity follows. Now we apply Lemma 15 to conclude that $C(\{1, \ldots, K\}, t) \leq \log_2(K+1)$ for all $t$, which implies $\mathbb{E}[N] \leq \log_2(K+1)$.

Finally, let $P$ denote the total number of phases. We can show that $P \leq 2 + \max(-1, \log_2(\min_i \mathcal{S}_i([1,T], \vec{c})))$. The proof of this claim is identical to that in Theorem 10 and is omitted for brevity. Let $N_p$ and $\gamma_p = 2^{p-1}$ denote the number of sub-phases in phase $p$ and the corresponding value for $\gamma$ respectively. We can then conclude the total expected regret experienced by Algorithm 4 is

$$\mathbb{E}\left[\sup_{u \in X} \sum_{t=1}^{T} \langle c_t, x_t - u \rangle\right] \leq \sum_{p=1}^{P} \mathbb{E}[N_p](\gamma_p + 1) \leq (2^P + P) \cdot \log_2(K+1)$$

$$\leq \log_2(K+1)\left(4 + 4 \min_i \mathcal{S}_i([1,T], \vec{c})\right).$$

To prove the second bound for an adaptive adversary, we simply observe that in the worst-case, we cannot have more than $K$ sub-phases in any phase. The rest of the argument is identical. □

In order to prove Theorem 11, we need the following technical Lemma:

**Lemma 15.** *Let $F : [K] \times [T] \to [T]$ be such that $F(i,t) \geq t$ for all $i \in [K], t \in [T]$ and $C : 2^{[K]} \times [T] \to \mathbb{R}$ be a function that satisfies $C(\emptyset, t) = 0$ for all $t$, $C(S, T) = 1$ for all $S$, $C(S, T+1) = 0$ for all $S$, and $C$ satisfies the recursion:*

$$C(S,t) = 1 + \frac{1}{|S|} \sum_{i \in S} C(S \setminus \{j \in S \mid F(j,t) \leq F(i,t)\}, F(i,t) + 1).$$

*Then $C(\{1, \ldots, K\}, t) \leq \log_2(K+1)$ for all $t$.*

*Proof.* We define the auxiliary function $Z(N) = \sup_{t, |S| \leq N} C(S,t)$. Observe $Z(0) = 0$, $Z(1) = 1$, and $Z(N)$ is non-decreasing with $N$. Now suppose for purposes of induction that $Z(n) \leq \log_2(n+1)$ for $n < N$. Then we have

$$Z(N) \leq 1 + \sup_{N' \leq N} \frac{1}{N'} \sup_{t, |S|=N'} \sum_{i \in S} C(S - \{j \in S \mid F(j,t) \leq F(i,t)\}, F(i,t) + 1)$$

$$\leq 1 + \sup_{N' \leq N} \frac{1}{N'} \sup_{t, |S|=N'} \sum_{i \in S} Z(N' - |\{j \in S \mid F(j,t) \leq F(i,t)\}|).$$

Now since $Z(n)$ is non-decreasing in $n$, this is bounded by:

$$\leq 1 + \sup_{N' \leq N} \frac{1}{N'} \sum_{i=1}^{N'} Z(N' - i)$$

$$\leq 1 + \sup_{N' \leq N} \frac{1}{N'} \sum_{i=1}^{N'} \log_2(N' - i + 1).$$

Now we apply Jensen inequality to the concave function $\log_2(n)$:

$$\leq 1 + \sup_{N' \leq N} \log_2 \left( \frac{1}{N'} \sum_{i=1}^{N'} N' - i + 1 \right)$$

$$\leq 1 + \sup_{N' \leq N} \log_2((N' + 1)/2)$$

$$= \log_2(N + 1).$$

To conclude, note that clearly $C(\{1, \ldots, K\}, t) \leq Z(K)$ for all $t$. $\qquad\square$

## E  Other applications of the combiner

In this section we discuss a couple of direct applications of our combiner algorithms to other settings.

### E.1  Adapting to different norms

For any $\ell_p$-norm, $p \in (1, 2]$, there is an algorithm that guarantees regret $\sup_{u \in \mathcal{B}} \frac{\|u\|_p}{\sqrt{p-1}} \sqrt{\sum_{t=1}^T \|c_t\|_q^2}$ where $q$ is such that $\frac{1}{p} + \frac{1}{q} = 1$ (such bounds can be obtained by e.g., the adaptive FTRL analysis described in [19], or see [24] for a non-adaptive version). However, it is not clear which $p$-norm yields the best regret guarantee until we have seen all the costs. Fortunately, these are monotone regret bounds, so by making a discrete grid of $O(\log d)$ $p$-norms in a $d$-dimensional space we can obtain the best of all these bounds in hindsight up to an additional factor of $\log d$ in the regret. Specifically:

**Theorem 16.** *Let* $K = \lfloor (\log d)/2 \rfloor$, *let* $q_0 = 2$ *and* $\frac{1}{q_i} = \frac{1}{q_{i-1}} - \frac{1}{\log d}$ *for* $i \leq K$. *Define* $p_i$ *by* $\frac{1}{q_i} + \frac{1}{p_i} = 1$. *For each* $i \in [K]$, *let* $\mathcal{A}_i$ *be an online learning algorithm that guarantees regret* $\sup_{u \in \mathcal{B}} \frac{\|u\|_{p_i}}{\sqrt{p_i - 1}} \sqrt{\sum_{t=1}^T \|c_t\|_{q_i}^2}$. *Then combining these algorithms using Algorithm 2 yields a worst-case regret bound of:*

$$\mathbb{E}[\mathcal{R}_{\mathcal{A}}(\mathcal{B}, \vec{c})] \leq O \left( (\log \log d) \cdot \inf_p \sup_{u \in \mathcal{B}} \frac{\|u\|_p}{\sqrt{p-1}} \sqrt{\sum_{t=1}^T \|c_t\|_q^2} \right).$$

### E.2  Simultaneous Adagrad and dimension-free bounds

The adaptive online gradient descent algorithm of [15] obtains the regret bound $D_2 \sqrt{\sum_{t=1}^T \|c_t\|_2^2}$, where $D_2$ is the $\ell_2$-diameter of $\mathcal{B}$. In contrast, the Adagrad algorithm obtains the bound $D_\infty \sum_{i=1}^d \sqrt{\sum_{t=1}^T c_{t,i}^2}$ where $D_\infty$ is the $\ell_\infty$-diameter of $\mathcal{B}$ and $c_{t,i}$ is the $i$th component of $c_t$ [10]. Adagrad's bound can be extremely good when the $c_t$ are sparse, but might be much worse than the adaptive online gradient descent bound otherwise. However, both bounds are clearly monotone, so by applying our combiner construction, we have:

**Theorem 17.** *There is an algorithm* $\mathcal{A}$ *such that for any sequence of vectors* $\vec{c}$, *the regret is at most:*

$$\mathbb{E}[\mathcal{R}_{\mathcal{A}}(\mathcal{B}, \vec{c})] \leq O \left( \min \left\{ D_2 \sqrt{\sum_{t=1}^T \|c_t\|_2^2}, \ D_\infty \sum_{i=1}^d \sqrt{\sum_{t=1}^T c_{t,i}^2} \right\} \right).$$

# F  Proof of Theorem 13

**Theorem 13.** *There is an algorithm $\mathcal{A}$ for the unconstrained setting such that for any $u \in \mathbb{R}^d$ and any $\alpha \in (0,1)$, we have*

$$\mathcal{R}_{\mathcal{A}}(u, \vec{c} \mid H) = O\left(\inf_{w \in \Delta_K} \left\{ \|u\|(\log T)\left(\frac{\sqrt{\log K}}{\alpha} + \sqrt{\frac{B_{\alpha}^{H(w)}}{\alpha}}\right)\right\}\right).$$

*Proof.* Algorithm $\mathcal{A}$ instantiates one $d$-dimensional *parameter-free* OLO algorithm $\mathcal{A}'$ that outputs $x_t$, gets costs $c_t$, and guarantees regret for some user specified $\epsilon$:

$$\sum_{t=1}^{T} \langle c_t, x_t - u \rangle \leq \epsilon + O\left(\|u\|\log(T) + \|u\|\sqrt{\sum_{t=1}^{T} \|c_t\|^2 \log \frac{T}{\epsilon}}\right).$$

Where the $O$ hides absolute constants. Such algorithms are described in several recent works [7, 8, 27, 17, 20]. Also, algorithm $\mathcal{A}$ instantiates $K$ one-dimensional learning algorithms, $\mathcal{A}_i$ for the hint sequence $\vec{h^{(i)}}$. At time $t$, the $i$th such learner outputs $y_t^{(i)}$, gets cost $-\langle c_t, h_t^{(i)}\rangle$ and guarantees regret:

$$\sum_{t=1}^{T} \langle c_t, h_t^{(i)}\rangle(y^{(i)} - y_t^{(i)}) \leq \frac{\epsilon}{K} + O\left(|y^{(i)}|\log(T) + |y^{(i)}|\sqrt{\sum_{t=1}^{T} \langle c_t, h_t^{(i)}\rangle^2 \log \frac{KT}{\epsilon}}\right)$$

$$\leq \frac{\epsilon}{K} + O\left(|y^{(i)}|\log(T) + |y^{(i)}|\sqrt{\sum_{t=1}^{T} \|c_t\|^2 \log \frac{KT}{\epsilon}}\right).$$

These one-dimensional learners may simply be instances of the $d$-dimensional learner restricted to one dimension. The algorithm $\mathcal{A}$ responds with the predictions $\hat{x}_t = x_t - \sum_{i=1}^{K} y_t^{(i)} h_t^{(i)}$ and set $\epsilon = 1$. The regret is:

$$\sum_{t=1}^{T} \langle c_t, \hat{x}_t - u \rangle = \sum_{t=1}^{T} \langle c_t, x_t - u \rangle - \sum_{i=1}^{K}\sum_{t=1}^{T} \langle c_t, h_t^{(i)}\rangle y_t^{(i)}$$

$$= \inf_{y^{(1)},\ldots,y^{(K)} \in \mathbb{R}} \left\{ \sum_{t=1}^{T} \langle c_t, x_t - u \rangle + \sum_{i=1}^{K}\sum_{t=1}^{T} \langle c_t, h_t^i\rangle(y^{(i)} - y_t^{(i)}) - \sum_{t=1}^{T}\left\langle c_t, \sum_{i=1}^{K} y^{(i)} h_t^{(i)}\right\rangle\right\}$$

$$\leq O\left(\inf_{y^{(1)},\ldots,y^{(K)} \in \mathbb{R}}\left\{1 + \|u\|\sqrt{\sum_{t=1}^{T}\|c_t\|^2 \log T} + \sum_{i=1}^{K}\left(\frac{1}{K} + |y^{(i)}|\sqrt{\sum_{t=1}^{T}\|c_t\|^2 \log(KT)}\right)\right.\right.$$

$$\left.\left.+\|u\|\log(T) + \sum_{i=1}^{K}|y^{(i)}|\log(T) - \sum_{t=1}^{T}\left\langle c_t, \sum_{i=1}^{K} y^{(i)} h_t^{(i)}\right\rangle\right\}\right)$$

$$\leq O\left(2 + \inf_{\sum_i |y^{(i)}| \leq \|u\|\sqrt{\frac{\log T}{\log(KT)}}}\left\{2\|u\|\log(T) + 2\|u\|\sqrt{\sum_{t=1}^{T}\|c_t\|^2 \log T} - \sum_{t=1}^{T}\left\langle c_t, \sum_{i=1}^{K} y^{(i)} h_t^{(i)}\right\rangle\right\}\right).$$

Let $w$ be an arbitrary element of $\Delta_K$. We set $y^{(i)} = \|u\|\frac{w^{(i)}}{\sqrt{\alpha|B_{\alpha}^{H(w)}| + \frac{\log(KT)}{\log T}}}$. Notice that this implies

$\sum |y^{(i)}| \leq \|u\|\sqrt{\frac{\log T}{\log(KT)}}$. Also, we have

$$-\sum_{t=1}^{T}\langle c_t, H(w)_t\rangle \leq -\sum_{t=1}^{T} \alpha\|c_t\|^2 + 2|B_{\alpha}^{H(w)}|, \quad \text{and}$$

$$-\sum_{t=1}^{T}\left\langle c_t, \sum_{i=1}^{K} y^{(i)} h_t^{(i)}\right\rangle \leq -\frac{\|u\|}{\sqrt{\alpha|B_{\alpha}^{H(w)}| + \frac{\log(KT)}{\log T}}}\sum_{t=1}^{T}\alpha\|c_t\|^2 + 2\|u\|\sqrt{\frac{|B_{\alpha}^{H(w)}|}{\alpha}}.$$

Thus the regret bound for $\mathcal{A}$ becomes

$$\mathcal{R}_{\mathcal{A}}(u, \vec{c} \mid H) \leq O \left( 2 + w\|u\| \log(T) + 2\|u\| \sqrt{\frac{|B_{\alpha}^{H(w)}|}{\alpha}} \right.$$

$$+ 2\|u\| \sqrt{\sum_{t=1}^{T} \|c_t\|^2 \log T} - \frac{\|u\|}{\sqrt{\alpha|B_{\alpha}^{H(w)}| + \frac{\log(KT)}{\log T}}} \left. \sum_{t=1}^{T} \alpha\|c_t\|^2 \right)$$

$$\leq O \left( 2 + \frac{\|u\|(\log T) \sqrt{\alpha|B_{\alpha}^{H(w)}| + \frac{\log(KT)}{\log T}}}{\alpha} + 2\|u\| \sqrt{\frac{|B_{\alpha}^{H(w)}|}{\alpha}} \right)$$

$$= O \left( \frac{\|u\| \sqrt{(\log T) \log(KT)}}{\alpha} + \|u\|(\log T) \sqrt{\frac{|B_{\alpha}^{H(w)}|}{\alpha}} \right).$$

Since $w$ was chosen arbitrarily in $\Delta_K$, the bound holds for all $w \in \Delta_K$ and so we are done. $\qquad\square$

## Footnotes

[1]All phases except maybe the last phase have exactly $K$ sub-phases.