[Reviews · NeurIPS 2020]

Review 1

Summary and Contributions: This paper follows a recent line of work on online learning with hints. In particular, the authors extend recent works considering a scenario where multiple hints are available at the beginning of any round in the game. The algorithms proposed by the authors can take advantage of the fact that the hints could be "well" correlated with the cost vectors, in order to reduce the final regret bound. On the other hand, in the worst-case (i.e. when the hints are not useful) the usual $ \sqrt{T} $-rate is recovered.

Strengths: The setting of online learning with many hints is not new (see for example section 5 of [9] "Combine Online Learning Guarantees" ). However, in this work the authors deal with imperfect hints and extend the setting recently introduced by [2] "Online Learning with Imperfect Hints". The authors provide different algorithms and carefully analyse their regret bounds. Furthermore the analysis is complemented with two different lower bounds. Finally, a new interesting technique for combining online learning algorithms is provided inspired by a clever use of the doubling trick, which might be of independent interest.

Weaknesses: I have no particular concerns about weaknesses. Probably the presentation of the proofs could be improved in some parts (see questions below). **After rebuttal**: the authors addressed all the questions asked. It seems that the usage of the smooth hinge loss is necessary as well as new combiner algorithms. Both are nice contribution that could be used beyond the scope of this paper.

Correctness: The theorems seem correct.

Clarity: The paper is well written and clear. However, the proof are sometimes not easy to follow.

Relation to Prior Work: Prior work is adequately discussed. On the other hand, it seems to me that the techniques given to combine online algorithms are inspired by algorithms from the theoretical computer science field (in particular the marking algorithm used for paging), together with a clever use of the doubling trick. However, none of the two are mentioned in the main paper.

Reproducibility: Yes

Additional Feedback: Some questions and suggestions for the authors: 1 - I would point out that the algorithm described in lines 155-160 is a version of the exponential weights algorithm, if I'm not wrong (you're using negative entropy regularisation after all). 2 - It is not totally clear to me why we need to use the smoothed hinge loss. As explained in section 3, we can cast the problem into the learning with expert advice setting. In this setting the multiplicative weights algorithms does not give the bound on a convex combination of experts. But then can't we use another experts' algorithm that gives the desired bound instead of all the machinery required for the smoothed hinge loss? In particular I'm thinking about Squint, which gives a second order bound and should be able to provide a first-order bound as the one used in your proofs. 3 - I would mention that the proof of proposition 3 is in the appendix. Also, the presentation of the proofs in propositions 3 need to be improved in my opinion: property "c" is not immediate from the definition; property "e" is a bit sketchy: you should assume that || c_t ||^2 is greater than 0, otherwise the first assertion is not “valid”. Finally, from the definition of $ \ell(a, b) $ I think that the $ \geq $ sign in property "b" should be just a $ > $ (and $ \leq $ instead of $ < $ in "c"). 4 - In the proof of thm 2 I think a sum over time is missing on the right hand side of eq. 6 5 - In proposition 4, I would say that FTRL is run with entropic regularisation and the learning rate is ~ 1 / ( sum of the gradients ), if I'm not wrong 6 - line 420, second inequality: are we assuming that $ \alpha < 1 $? 7 - line 421, the inequality holds for $ a \geq 0, X \geq 0 $. 8 - line 249: shouldn't it be $\gamma$ instead of $\lambda = 2^{p-1}$? 9 - Do you think you could get a high probability bound for the randomized combiner?


Review 2

Summary and Contributions: The paper studies the setting of online learning with hints. The paper generalizes previous results to the case where more than one hint is available at each round. The new regret bounds are small whenever there exists a fixed weighted combination of hints which correlate well with the sequence of cost vectors. When the decision set is bounded, the proposed algorithm uses a new "experts" subroutine for linear costs whose regret is a multiplicative factor away from the best expert's regret. This result of independent interest.

Strengths: The paper makes a nice contribution to the setting of online linear optimization with hints. The paper builds on the work Bhaskara et al. 2020 but focus more on the setting where the decision set is bounded. The algorithmic ideas are simple and easy to follow. The dependence in the correlation level \alpha in the regret bound's main term is improved compared to previous results---\alpha now appears inside the square-root multiplying the main term of the bound. The paper also presents a new expert algorithm for linear costs whose regret is less than a constant times the regret of the best expert in hindsight. This is also a nice contribution.

Weaknesses: The 'expert' algorithm of section 4 isn't really discussed in light of existing results (e.g. algorithms with small loss bounds). The paper does mention the work of Cutkosky 2019 on 'Combining Online Learning Guarantees' but fails to discuss the similarities/differences between the obtained bounds. The regret bound of Cutkosky 2019 is better because there is no multiplicative factor greater than one multiplying the regret of the best expert learner on the RHS. However, their bound only holds for parameter-free online learners. The authors of the current paper claim that the result of Cutkosky 2019 only holds for the bounded decision set setting, which provides the ground to introduce their new expert (or combiner) algorithm. This claim is, however, false. The restriction on the result of Cutkosky 2019 is that the base learners (or experts) need to be parameter-free in that their regret bounds need to scale with the norm of the chosen comparator. Such algorithms are also available for the bounded setting via an existing reduction; see 'Artificial constraints and hints for unbounded online learning.' This makes me wonder if the expert algorithm of Cutkosky 2019 could have been used in place of the new "combiner algorithm"; if the base algorithms in Theorem 10 can be made "parameter-free", then this is certainly possible. Applying Cutkosky's algorithm would also avoid multiplying by the number of experts in the regret bounds e.g. Theorem 12. That being said, even if it turns out that the new combiner algorithm was not needed after all for the goal set out by this paper, the new combiner algorithm might be useful in other cases where the experts are not necessarily parameter-free. But I think this should be discussed in the paper. Other less severe weaknesses include missing quantifiers in the statements of some theorems. One example is Theorem 13, which should hold for all \alpha and the comparator u but this is not mentioned. Also, I find it somewhat non-standard to write the regret bounds without specifying the choice of comparator; in Theorem 13, again, the comparator appears on the RHS but not on the LHS. I would suggest considering changing the notation for the regret to include the choice of the comparator or finding a way to make this choice clearer. Finally, there still seems to be a gap between the lower bounds in section 3.3 and the regret bounds presented in the paper. For instance, it is shown in Theorem 7 that a \sqrt{\log K/ \alpha} is unavoidable, but the algorithms of the paper do not achieve this; in the bound of Theorem 5, there is a \sqrt(\ln T \ln K)/\alpha term which is worse than \sqrt{\log K/ \alpha}. The authors do not discuss this gap in the paper. In general, more discussion could go in subsection 3.3.

Correctness: I went through most of the proofs. The claims seem correct.

Clarity: Writing and discussions could be improved in some places. But this would not require significant changes.

Relation to Prior Work: In general yes, except perhaps for the new combiner algorithm---see my comments under the "weaknesses" section.

Reproducibility: Yes

Additional Feedback:


Review 3

Summary and Contributions: This is a theoretical paper studying online linear optimization in the presence of many imperfect hints. The difficulty of this setting is that the benefit of aggregating things is a non-linear function of the individual hints. The main technical contribution of the paper is to provide a new algorithm that obtains a regret bound that is essentially tight. The results highlight that the algorithm can compete with the best convex combination of the hints. The algorithm is designed using a combination a smoothed hinge loss with Mirror Descent.

Strengths: The paper is clearly relevant to NeurIPS and contributes in a clear way to the literature on OLO. The results are nearly tight and the algorithm is simple and intuitive.

Weaknesses: I worry that the paper is of niche interest. While OLO is a classical area, it is well-trodden ground and it is not clear that the algorithmic ideas or analytic technique extend beyond the narrow context of the paper. Additionally, while the result is clearly novel, it is unclear how novel the technical insights that lead to the results are. I would like to ask the authors to comment on the novelty of the proof technique during the response.

Correctness: I have read the proofs and did not find any issues

Clarity: The paper is clear but could do a better job of providing broader context. In particular, the relation of the algorithm and proof techniques to the related work is not made clear to me. Additionally, more discussion of the qualitative insights provided by the results would improve the paper.

Relation to Prior Work: The relationship to prior work is discussed only briefly in the introduction. The paper would be improved if this was extended and the results in the paper were placed more clearly in the context of the related papers described in the introduction.

Reproducibility: Yes

Additional Feedback: Thank you for your author response. Based on reading it and given discussions with the other reviewers I have increased my assessment of the paper. I am less worried about the potential for "niche" interest given that it does seem that the techniques will be applicable more broadly. I encourage you to incorporate some of the discussions of the techniques and the applications in the appendix into the body if the paper is accepted.

[Author Response · NeurIPS 2020]

We thank all the reviewers for their detailed comments and suggestions.

**Review #2:**

1: Thanks for pointing this connection. Yes, it is a version of the exponential weights, where the losses are the gradients
of the smoothed hinge loss. We will add this remark to the revision.

2: The smoothed loss is actually quite important to our algorithm. While one could likely use an algorithm with a
second-order bound like Squint to learn the weights instead of our FTRL-based approach (which we chose mainly for
simplicity), we believe that the losses provided to the algorithm must be the gradients of the *smoothed loss* rather than
just the individual correlations of the hints. The reason we think the smoothed loss is necessary (as discussed in lines
127-132) is that a linear combination of individually bad hints might produce a hint that is actually extremely good. It is
not obvious how to capture this without using a loss that aggregates information from all hint sequences as our smooth
loss does.

3: We agree with the suggestions and will incorporate them; thank you for the careful reading.

4, 5, 7, 8: We apologize for these typos and slight inaccuracies; we will fix them in the revision.

6: Yes, we assume $\alpha \leq 1$.

9: Our current proof is unable to show a high probability bound, but this is an interesting question, thanks! We will add
this remark to the revision. We will also add the similarity of our analysis to quick-sort and paging.

**Review #3:**

Regarding the combiner algorithm and comparison to [Cutkosky 2019]: at first blush it is true that the [Cutkosky
2019] combiner only requires small loss at the origin, but it is actually difficult to use it in constrained settings even
when appealing to the black-box constraint set reduction proposed in [Cutkosky & Orabona 2018]. This is because the
reduction changes the losses in a way that might damage the regret bounds of the algorithms that are being combined. In
particular, the reduction requires one to commit to a particular norm, which makes it difficult to design an algorithm that
combines base algorithms that use different $p$ norms and is also constrained, as we are able to accomplish in Theorem
16. In fact, even in [Cutkosky 2019], the constrained optimism algorithm requires an ad hoc technical modification to
the constraint set reduction in order to work. Nevertheless, as you suggest, we will add a short discussion about these
subtleties and the limitations of the prior work.

Quantifiers: We will add the missing quantifiers to theorem statements to make them clearer. As suggested, we will add
the comparator $u$ to the notation for regret in all appropriate places.

As you correctly notice, there is a small gap of $\sqrt{(\ln T)/\alpha}$ in the bounds. We will add a discussion to this effect.

**Review #4:**

Our main algorithm has two parts: (i) an algorithm to find an optimal combination of different hint sequences for a
known value of $\alpha$, and (ii) a combiner algorithm to deal with unknown $\alpha$. To the best of our knowledge, both involve
novel techniques that potentially have broader applications.

(i) Smooth hinge loss: Since a combination of multiple hint sequences can be significantly superior to any of the
individual hint sequences, using the individual correlations of the hints is not sufficient. We introduce a novel smoothed
hinge loss precisely to deal with this issue and show that using FTRL on these new losses helps obtain a new hint
sequence that can provide regret comparable to the best combination of the original hints.

(ii) Combiner algorithm: This is a new, general way to combine $K$ online learners while obtaining regret that is as
good as that of the best learner, to a factor $\log K$. The closest work we are aware of is [Cutkosky 2019, "Combining
Online Learning Guarantees"], but our approach is conceptually quite different and applies in different settings (see our
response to Review 3 for discussion of the differences). Furthermore, we show how the combiner can be used to obtain
new results outside the setting of our problem. Appendix E contains two such applications: adapting to different norms
and simultaneous Adagrad and dimension-free bounds.

In addition to adding these remarks, in the revision, we will position our algorithms and the proof techniques better
with respect to related literature.

[Meta-Review · NeurIPS 2020]

All reviewers agree on the theoretical contribution of this work. In addition to the discussions that the author promised to add in the rebuttal, it would be great if more examples are included in the final version to motivate where the hints might be from.